# The 2020 “Padua Criteria” for Diagnosis and Phenotype Characterization of Arrhythmogenic Cardiomyopathy in Clinical Practice

**DOI:** 10.3390/jcm11010279

**Published:** 2022-01-05

**Authors:** Francesca Graziano, Alessandro Zorzi, Alberto Cipriani, Manuel De Lazzari, Barbara Bauce, Ilaria Rigato, Giulia Brunetti, Kalliopi Pilichou, Cristina Basso, Martina Perazzolo Marra, Domenico Corrado

**Affiliations:** Department of Cardiac, Thoracic and Vascular Sciences, University of Padua, Via Giustiniani 2, 35128 Padova, Italy; francesca.graziano.1@studenti.unipd.it (F.G.); alessandro.zorzi@unipd.it (A.Z.); alberto.cipriani@unipd.it (A.C.); manuel.delazzari@aopd.veneto.it (M.D.L.); barbara.bauce@unipd.it (B.B.); ilaria.rigato@aopd.veneto.it (I.R.); giulia.brunetti@studenti.unipd.it (G.B.); kalliopi.pilichou@unipd.it (K.P.); cristina.basso@unipd.it (C.B.); martina.perazzolomarra@unipd.it (M.P.M.)

**Keywords:** arrhythmogenic cardiomyopathy, cardiac magnetic resonance, cardiomyopathy, diagnosis, ventricular arrhythmias

## Abstract

Arrhythmogenic Cardiomyopathy (ACM) is a heredo-familial cardiac disease characterized by fibro-fatty myocardial replacement and increased risk of sudden cardiac death. The diagnosis of ACM can be challenging due to the lack of a single gold-standard test: for this reason, it is required to satisfy a combination of multiple criteria from different categories including ventricular morpho-functional abnormalities, repolarization and depolarization ECG changes, ventricular arrhythmias, tissue characterization findings and positive family history/molecular genetics. The first diagnostic criteria were published by an International Task Force (ITF) of experts in 1994 and revised in 2010 with the aim to increase sensitivity for early diagnosis. Limitations of the 2010 ITF criteria include the absence of specific criteria for left ventricle (LV) involvement and the limited role of cardiac magnetic resonance (CMR) as the use of the late gadolinium enhancement technique for tissue characterization was not considered. In 2020, new diagnostic criteria (“the Padua criteria”) were proposed. The traditional organization in six categories of major/minor criteria was maintained. The criteria for identifying the right ventricular involvement were modified and a specific set of criteria for identifying LV involvement was created. Depending on the combination of criteria for right and LV involvement, a diagnosis of classic (right dominant) ACM, biventricular ACM or left-dominant ACM is then made. The article reviews the rationale of the Padua criteria, summarizes the main modifications compared to the previous 2010 ITF criteria and provides three examples of the application of the Padua criteria in clinical practice.

## 1. Background

Arrhythmogenic cardiomyopathy (ACM) is an inherited heart muscle disease characterized by progressive fibro-fatty replacement and malignant ventricular arrhythmias that may lead to sudden cardiac death, especially in young people and athletes [1,2]. The disease was originally considered a development defect of the right ventricular (RV) myocardium, so it was called “dysplasia” [3]. The discovery of genetic defects in genes encoding cardiac desmosomes [4] led to a more appropriate definition of arrhythmogenic right ventricular “cardiomyopathy” (ARVC). Thus, ARVC was definitively introduced in the World Health Organization classification of cardiomyopathies [5]. Subsequently, a broader definition of “arrhythmogenic cardiomyopathy” (ACM) was introduced as a consequence of the identification of biventricular and left-dominant variants [4,6].

The diagnosis of ACM is challenging due to the lack of a single sensitive and specific test. In 1994 an International Task Force (ITF) of experts in cardiomyopathies proposed the first diagnostic criteria based on a multiparametric approach. These criteria relied on the identification of different clinical abnormalities typical of the disease: dilation/dysfunction of the RV; ECG changes; ventricular arrhythmias; histopathological abnormalities and positive family history [7]. Each category included “major” and “minor” criteria based on their specificity for ARVC. The diagnosis was reached either with two major criteria, or one major plus two minor criteria, or four minor criteria. The perspective at that time was that ACM was a predominantly RV disease, with involvement of the left ventricle (LV) only in the late stages. The limits of the 1994 ITF criteria were the lack of quantitative parameters and the low sensitivity for early phenotypes [8]. As a result, in 2010 the Revised ITF criteria were published [9]. They were enriched with quantitative imaging and histomorphometry reference values and the categories of ECG and ventricular arrhythmias were updated. Moreover, the positive genetic testing for desmosomal mutation was added to the family history category. Finally, in addition to the threshold for the definite diagnosis that remained unchanged, the diagnosis was considered possible when two minor criteria or one major criterion only were satisfied and borderline with three minor criteria or one major plus one minor criteria.

In 2019 an international experts’ report provided an extensive critical appraisal of the 2010 ITF criteria, identifying potential areas of improvement [10]. The major limitations of the 2010 ITF criteria that were pointed out were the lack of specific criteria for diagnosis of the broader phenotypic spectrum of the disease, which includes left-sided variants, and the lack of tissue characterization findings by cardiac magnetic resonance (CMR) using the late gadolinium enhancement (LGE) technique. Inclusion of the LGE among the diagnostic criteria is essential for diagnosing LV involvement that can be characterized by subepicardial fibrous or fibro-fatty scars without associated ventricular wall motion abnormalities. Based on this report, in 2020, an international expert consensus document provided upgraded diagnostic criteria for ACM (the “Padua Criteria”) [11]. 

## 2. The Padua Criteria for ACM Diagnosis

The Padua criteria are organized in two different sets of criteria to identify, respectively, clinical signs of RV and LV involvement. In both sets of criteria, the traditional organization in six diagnostic categories is maintained, including morpho-functional changes, tissue characterization, repolarization and depolarization ECG abnormalities, ventricular arrhythmias and family history/genetic testing. Similar to the 2010 ITF criteria, the diagnostic criteria are divided into “major” and “minor” and the diagnosis is considered possible, borderline or definite according to the number of criteria that are fulfilled. However, to reach the diagnosis of ACM, at least one morpho-functional or structural criteria (either major or minor) needs to be satisfied. (Table 1).

The use of the 2020 International criteria is a two-step process: the first step is the application of the multiparametric approach to verify how many major/minor criteria for both RV and LV involvement are satisfied. It is noteworthy than only one major or minor criterion for each category can be considered. The second is to classify the ACM phenotype in one of three different variants (classic right dominant ARVC, biventricular ACM or LV arrhythmogenic cardiomyopathy, ALVC) according to the combination of criteria (Figure 1).

### 2.1. STEP 1: The Multiparametric Diagnostic Approach

1. Morpho-functional abnormalities

The morpho-functional abnormalities can be detected with echocardiography, CMR or angiography. At variance with the 2010 ITF criteria, the presence of RV wall motion abnormalities (akinesia, dyskinesia or bulging) without associated global RV dilation/dysfunction is now classified as a minor criterion. Despite its high specificity for ARVC [13], isolated RV regional abnormalities are considered a minor criterion because of the possible flaws resulting from a subjective assessment of RV kinetic, also deriving from potential misinterpretation of some normal variants of the RV wall motion [14]. If these alterations are associated with chamber dilatation or dysfunction, regardless of their severity, they are considered as a major criterion. 

The LV morpho-functional criteria include the presence of global systolic dysfunction (reduction of ejection fraction or of echocardiographic global longitudinal strain) with or without LV dilatation, or the documentation of regional hypokinesia or akinesia. They are both considered minor criteria because of the low specificity for ACM. The use of current reference values for cardiac chamber size and function (normalized for sex, age, body surface area) [15,16], and specific reference values for athletes [17], are recommended.

2. Structural myocardial abnormalities

The structural myocardial abnormalities are detected through CMR or endomyocardial biopsy (EMB). The major CMR criteria are the presence of transmural LGE in at least 1 RV region, and the presence of a stria of LGE with a non-ischemic distribution (subepicardial or midmyocardial) affecting at least 1 LV Bull’s Eye segment (excluding septal junctional LGE). In both cases, the LGE/fibrosis must be confirmed in two orthogonal plans. The “ring pattern” is a circumferential distribution of subepicardial LGE in the LV free wall and septum, seen in short axis view: it is highly specific for ALVC [18]. 

Right ventricular LGE has a high diagnostic specificity but low sensitivity due to the thin RV wall and the suboptimal resolution obtained with CMR. The combination of LGE and wall motion abnormalities results in the highest accuracy [19]. 

The fatty tissue replacement can be detected with dedicated sequences by CMR, and it is often observed in the same regions of LGE: however, it is not considered a diagnostic criterion when found in isolation because of its lack of specificity. 

The histological tissue characterization through EMB is indicated in patients with non-familial ACM and negative genotyping to exclude phenocopies (sarcoidosis, myocarditis or dilated cardiomyopathy) [4]. The demonstration of fibrous replacement of the RV with EMB, with or without fatty tissue, is a major structural criterion.

3. ECG repolarization abnormalities

Among the repolarization abnormalities, T wave inversion (TWI) in right precordial leads (V1–V3) or beyond is a major criterion for RV involvement. Instead, the presence of TWI only in V1–V2 is a minor criterion. The criteria are valid in the absence of complete right bundle branch block (RBBB) and in patients who have already achieved complete pubertal development. In case RBBB is present, TWI extension through V1–V4 is a minor criterion, as long as pubertal development is completed. TWI extending from V1 to V5 or V6 is the expression of a more severe RV dilatation, caused by its displacement to lateral leads, rather than of concomitant LV disease [20].

Only the presence of TWI in left precordial leads (V4–V6) in the absence of LBBB is a minor criterion for LV involvement.

4. ECG depolarization abnormalities

There are no major criteria among the depolarization abnormalities. The 2020 Padua criteria downgraded the epsilon wave in right precordial leads to minor criteria because it is largely influenced by ECG sampling rate and filtering, with a high interobserver variability [21]. In right precordial leads, a terminal activation duration of the QRS ≥ 55 ms from the nadir of the S wave to the end of the QRS without a complete RBBB is a minor ECG criterion, especially if followed by TWI. Signal averaged ECG is no longer considered, given the low diagnostic accuracy.

The fibro-fatty replacement involving the LV could be responsible for low QRS voltages in limb leads (<0.5 mV in all limb leads), which is a minor criterion in the absence of other potential causes (emphysema, pericardial effusion or obesity). Inappropriate setting (<100 Hz) of low band-pass filters can cause spurious QRS voltage attenuation.

5. Ventricular arrhythmias

Ventricular arrhythmias are typical of ACM and arise from or around the fibro-fatty tissue. Premature ventricular beats (PVBs) are considered in terms of absolute number (>500 PVBs per 24 h), complexity (non-sustained or sustained ventricular tachycardia, VT) and morphology on 12-ECG leads (exercise test or 24 h Holter monitoring). PVBs or VT with LBBB/superior axis morphology are more specific for ACM as they originate from the RV free wall or interventricular septum (major criterion). Instead, ventricular arrhythmias with an LBBB/inferior axis morphology are less specific (minor criterion), given that they originate from the RV outflow tract and are often idiopathic. 

PVBs or VT with an RBBB morphology, excluding the fascicular pattern (QRS < 130 ms), origin from the LV and are a minor LV criterion. The most common PVBs morphology in patients with an LV scar involving the lateral wall or infero-lateral wall, as typically observed in patients with biventricular ACM or ALVC, is RBBB/wide QRS/superior axis.

6. Family history and molecular genetics.

This category is shared by RV and LV criteria because it is not useful for phenotype characterization: in fact, the manifestation of the disease and the predominant involvement of one or the other ventricle may vary among members of the same family and in individuals with the same gene mutation.

The history of a first-degree relative with ACM confirmed pathologically at autopsy or surgery, or who received a diagnosis of definite ACM, is a major criterion. A minor criterion is met if the disease is confirmed in a second-degree relative, it is suspected but not confirmed in a first-degree relative or it is suspected in a first-degree relative who died suddenly at young age (<35 years old). 

Furthermore, the Padua criteria recommend genotyping in probands who satisfy the diagnosis of ARVC or biventricular ACM, to detect genetically affected family members at a preclinical phase of the disease. Genotyping may also be considered in borderline phenotypic patients to achieve the diagnosis (taking into account the current limitations of molecular genetic testing), and it is necessary to reach a diagnosis of purely LV ACM (i.e., ALVC) to exclude phenocopies such as ventricular scars from previous myocarditis [10].

### 2.2. STEP 2: The Phenotype Characterization

After a careful evaluation of the six diagnostic categories, the second step is to define the specific phenotypic variants and the likelihood of the disease (Figure 1). First, to reach any diagnosis of ACM, at least one major or minor morpho-functional or structural criteria from either the RV and LV criteria must be fulfilled.

If morpho-functional and/or structural criteria are met for both ventricles, either major or minor, the patient is diagnosed with biventricular ACM that can be considered definite, borderline or possible according to the number of additional criteria that are satisfied from both the LV and RV categories.

If morpho-functional and/or structural criteria are met only for the RV, the patient may be diagnosed with classic ARVC if a sufficient number of additional RV criteria are satisfied. In this case, the “electrical” LV criteria (ECG changes and ventricular arrhythmias with RBBB morphology) are not considered.

Finally, if no morpho-functional and/or structural RV criteria is satisfied, the diagnosis of ALVC requires the combination of the structural LV criterion (i.e., non-ischemic LV LGE) and a positive molecular genetic testing for ACM mutation.

Three examples of practical application of Padua criteria follow.

## 3. Practical Application of the Padua Criteria

### 3.1. Example 1

A 38-year-old woman was admitted for sustained VT. She reported two full-term pregnancies and that her mother suffered form an unspecified cardiac disease characterized by arrhythmias, no other relevant medical history. Physical examination was normal. ECG revealed TWI in V1–V5 and flattened T wave in inferior leads (Figure 2a). Exercise testing demonstrated frequent PVBs and a non-sustained VT with LBBB/superior axis morphology consistent with RV free wall origin (Figure 2b). Echocardiography demonstrated a moderate RV dilatation (EDV 15,57 cm^2^/m^2^), with hypokinesis of the free RV wall and lower normal limit (FAC 33%) RV function. CMR revealed a mild RV dilatation, a moderate systolic function reduction (EDV 116 mL/m^2^, EF 41%), a wide peri-tricuspid aneurysm, with an extreme thinning of the wall and apical hypertrabeculation (Figure 2c–f). Perhaps because of the thinning of the RV wall, no RV LGE was visualized. Genetic testing was proposed, but the patient decided to take time to consider it. Therefore, the patient achieved a RV major morpho-functional criterion, a major repolarization criterion, a major ventricular arrhythmias criterion and a minor family history criterion (Figure 2g). No LV criteria were satisfied. According to the Padua criteria, the final diagnosis was “definite ARVC” (Figure 2h).

### 3.2. Example 2

A 39-year-old man came to the emergency room complaining about oppressive chest pain. When he was 14, he had tuberculosis; he was an ex-smoker and he had hyperhomocysteinemia. The physical examination was normal. The ECG revealed low-voltage of QRS complex in the limb leads, ST-segment elevation in lateral leads and ST-segment depression in inferior leads. As an ST elevation myocardial infarction (STEMI) was suspected, he underwent emergent coronary angiography that demonstrated normal coronary arteries. The echocardiography revealed mild RV dilatation (EDV 14 cm^2^/m^2^) with normal function and subtricuspidal hypokinesis. The troponin peak was 177.600 ng/L (n.v. 0–34 ng/L). CMR was consistent with acute biventricular myocarditis. LV function was mildly reduced (EF 51%) with focal hypokinesis of the mid-lateral wall. There was also a mild reduction of the RV function (EF 42%) with subtricuspidal hypokinesis, right ventricle outflow tract (RVOT) and costophrenic angle bulging. Myocardial edema and LGE was present in both the LV, with a subepicardial distribution (non-ischemic pattern), and the RV. A more detailed collection of anamnestic data revealed that his maternal aunt suddenly died at the age of 50, and that some time before the patient had undergone a 24 h ECG Holter monitoring because of palpitations, which revealed infrequent isolated PVBs with RBBB/superior axis morphology. Therefore, he underwent an EMB that showed fibro-fatty replacement associated with acute inflammation consistent with “hot phase” ACM [22].

At six months follow-up, the ECG presented low QRS voltages in limb leads and flattened T wave in infero-lateral leads (Figure 3a). The stress test revealed polymorphic PVBs, the prevailing morphology was LBBB with negative precordial concordance/right axis deviation (Figure 3b). No frequent PVBs or non-sustained VT on 24-h ECG monitoring were recorded: for this reason, the arrhythmic criteria were not satisfied. The CMR was repeated, showing the disappearance of myocardial edema and confirming the presence of extensive LGE of both the LV, with a subepicardial/midmyocardial “ring-like” pattern, and of the RV. Fatty infiltration of the LV infero-basal and mid-lateral wall and of the RV free wall was also noted (Figure 3c–f). Finally, the genetic test revealed a homozygous truncation mutation, c.1660C>T (p.Q554X) in Desmocollin-2 (DSC2) gene. Therefore, according to the Padua criteria, a diagnosis of “definite biventricular ACM” was made (Figure 3g,h).

### 3.3. Example 3

A 23-year-old female was referred to the outpatient clinic for recurrent syncope and family history of ACM. Her paternal uncle died suddenly at the age of 28. Her father had a sudden cardiac arrest at the age of 45, then he was diagnosed with ACM, and he underwent implantable cardioverter defibrillator implantation. Her father’s genetic testing demonstrated the M1299GfsX6 mutation in Desmoplakin (DSP) gene, and the S596L mutation in Junction plakoglobin (JUP) gene. Physical examination was unremarkable. The ECG (Figure 4a) showed TWI in V1–V2, and terminal activation duration of QRS ≥ 55 ms. The exercise testing demonstrated PVBs with an RBBB/superior axis morphology (Figure 4b). No frequent PVBs or non-sustained VT on 24-h ECG monitoring were recorded: for this reason, the arrhythmic criteria were not satisfied.

The echocardiography was normal. CMR revealed normal chamber size and function, subepicardial/midmyocardial LGE of the LV, with a “ring-like” pattern, without any sign of RV involvement (Figure 4c–f). She underwent genetic testing. The exome 23 of the DSP gene and the exome 11 of the JUP gene were sequenced. The same pathogenic genetic variants of her father were demonstrated. According to the Padua criteria, the diagnosis was “ALVC” because of the absence of RV morpho-functional and structural criteria, and the presence of LV structural major criterion plus ACM-gene mutations (Figure 4g,h).

## 4. Conclusions

In conclusion, we provided an overview of the 2020 Padua criteria for ACM diagnosis and examples of their practical application. These criteria aim to improve the diagnosis of ACM, particularly by identifying LV involvement. The main element of novelty compared to the 2010 ITF criteria is the central role of CMR, which has become mandatory to characterize the ACM phenotype and to exclude other diagnoses. We believe that the application of the Padua criteria in clinical practice will be crucial for their validation, correlation with therapeutic outcomes and future refinement.

## Figures and Tables

**Figure 1 jcm-11-00279-f001:**
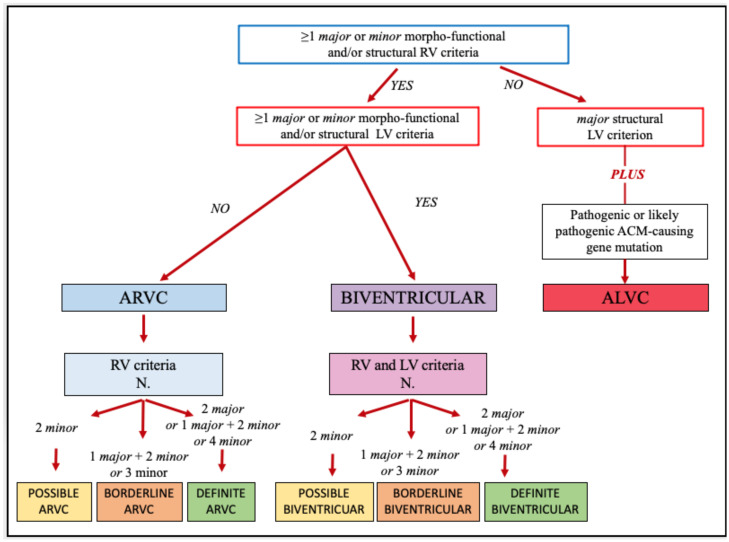
Flowchart for phenotypic characterization of arrhythmogenic cardiomyopathy [12].

**Figure 2 jcm-11-00279-f002:**
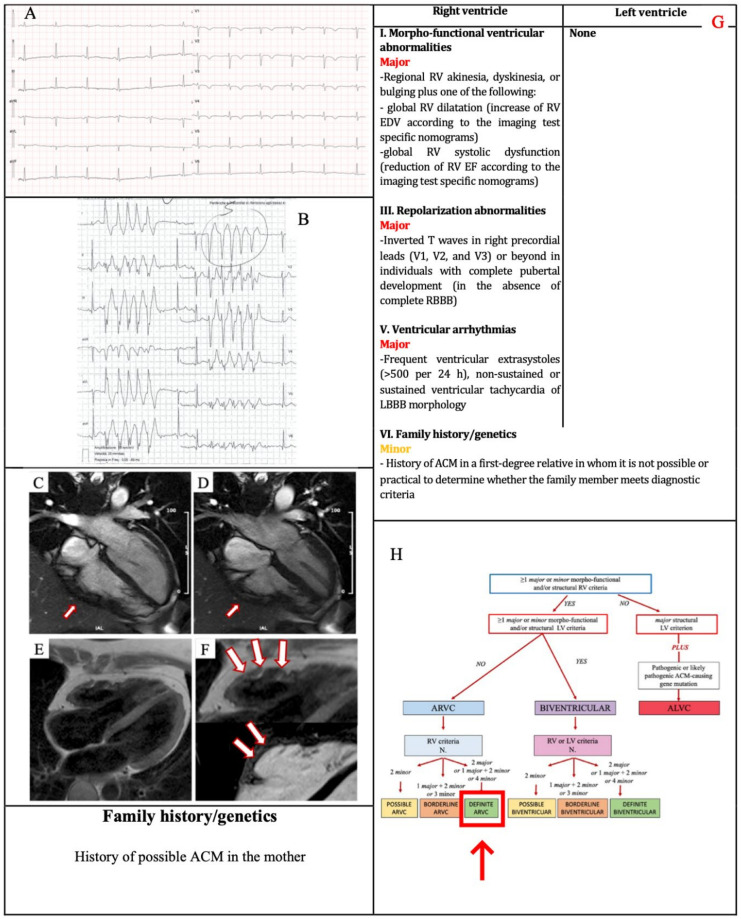
Example 1. (**A**) ECG. TWI in V1–V5 and flattened T wave in inferior leads. (**B**) Exercise testing. NSVT with LBBB/superior axis morphology. CMR. Four-chamber cine view in diastolic phase (**C**) and systolic phase, (**D**)*:* a wide peri-tricuspid aneurysm, with an extreme thinning of the wall (arrows). PD-TSE four-chamber view for fat evaluation, *(***E**)*:* fatty infiltration of the RV wall, in particular in the subtricuspid region (magnified on the top of **F**, arrows) On the corresponding postcontrast sequences (**F** on the bottom, arrows) the presence of RV LGE in the same region of fatty infiltration could not be evidenced. The 2020 Padua criteria achieved by the patient (**G**). Diagnostic flowchart for ACM phenotypic variants [12] (**H**). In the red box, the diagnosis of the patient. ACM = arrhythmogenic cardiomyopathy; CMR = cardiac magnetic resonance; LBBB = right bundle branch block; LGE = late gadolinium enhancement; NSVT = non-sustained ventricular tachycardia; PD–TSE = proton density-weighted turbo spin-echo; RV = right ventricle; TWI = T wave inversion.

**Figure 3 jcm-11-00279-f003:**
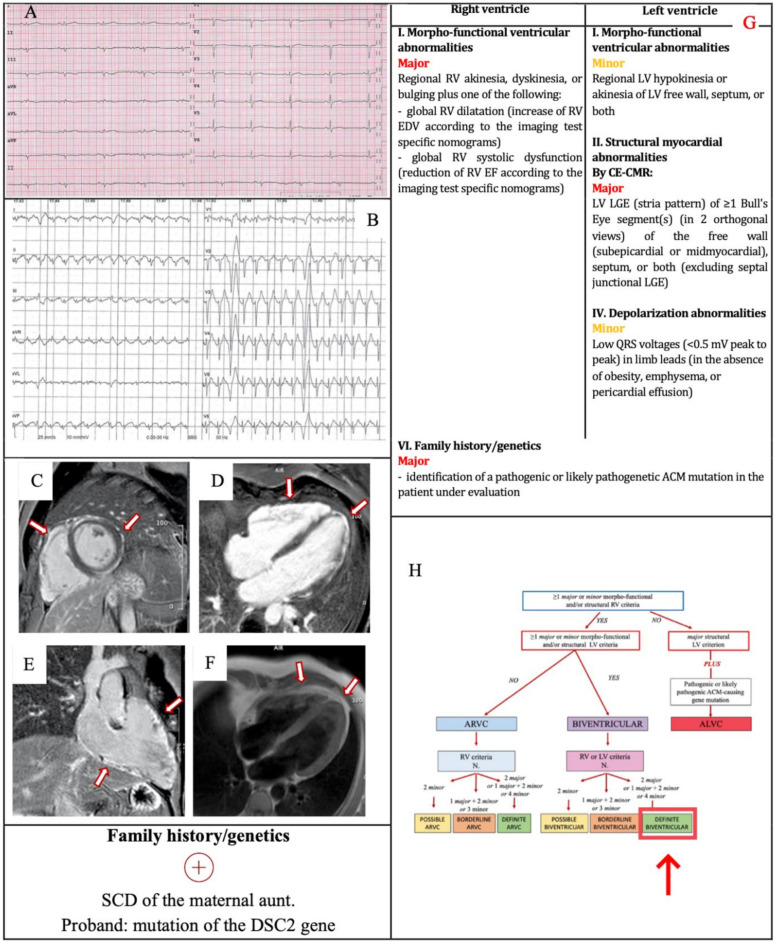
Example 2. (**A**) ECG. Low QRS voltages in limb leads and flattened T wave in infero-lateral leads. (**B**) Exercise testing. 2 PVBs with LBBB with negative precordial concordance/right axis deviation morphology. CMR. Postcontrast phase, short axis view (**C**), four-chamber view (**D**), RV inflow–outflow view (**E**): LGE of LV and RV (arrows). PD–TSE four-chamber view (**F**): fatty infiltration (arrows). The 2020 Padua criteria achieved by the patient (**G**). Diagnostic flowchart for ACM phenotypic variants [12] (**H**). In the red box, the diagnosis of the patient. CMR = cardiac magnetic resonance; DSC2 = Desmocollin-2; LBBB = left bundle branch block; LGE = late gadolinium enhancement; LV = left ventricle; PD–TSE = proton density-weighted turbo spin-echo; PVBs = premature ventricular beats; RV = right ventricle; SCD = sudden cardiac death.

**Figure 4 jcm-11-00279-f004:**
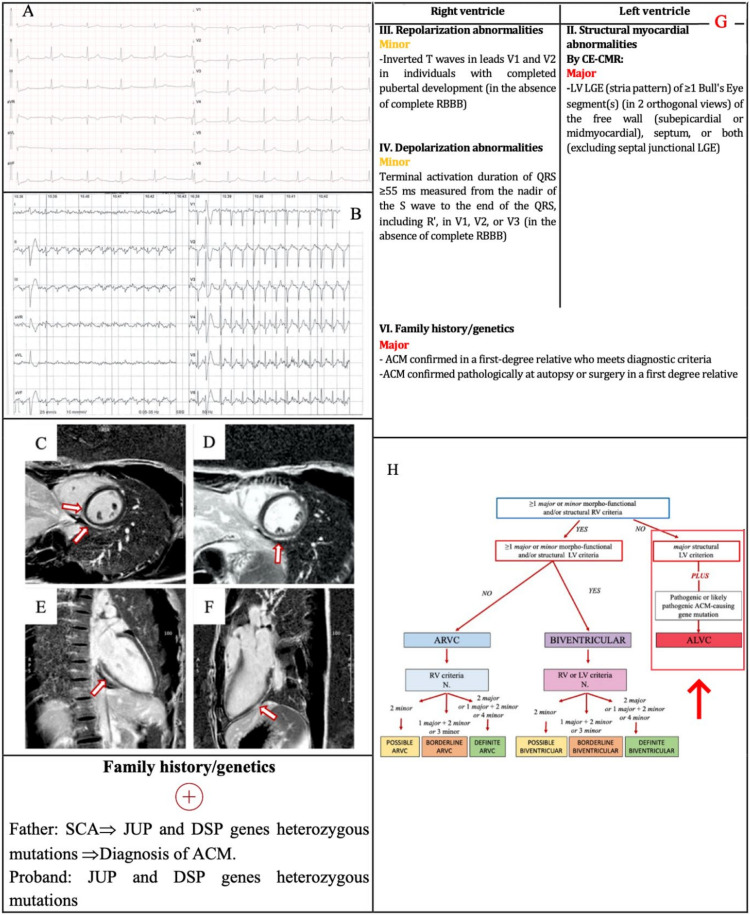
Example 3. (**A**) ECG. TWI in V1–V2 and terminal activation duration of QRS ≥ 55 ms in V1–V2. (**B**) Exercise testing. PVB with RBBB/superior axis morphology. CMR. Postcontrast phase, basal (**C**) and mid (**D**) short axis view, two-chamber view (**E**) and three-chamber view, (**F**): subepicardial/midmyocardial LGE of LV, in the inferior wall involving the adjacent interventricular septum on right side (**C**,**D**, arrows), with “ring-like” pattern in short axis mostly in the mid portion (**D**), confirmed by the orthogonal view (**E**,**F**). The 2020 Padua criteria achieved by the patient (**G**). Diagnostic flowchart for ACM phenotypic variants [12] (**H**). In the red box, the diagnosis of the patient. ACM = arrhythmogenic cardiomyopathy; CMR = cardiac magnetic resonance; DSP = Desmoplakin; JUP = Junction plakoglobin; LGE = late gadolinium enhancement; LV = left ventricle; PVB = premature ventricular beat; RBBB = right bundle branch block; SCA = sudden cardiac arrest; TWI = T wave inversion.

**Table 1 jcm-11-00279-t001:** The “Padua criteria”–ACM, arrhythmogenic cardiomyopathy; ALVC, arrhythmogenic left ventricular cardiomyopathy; BSA, body surface area; CECMR, contrast-enhanced cardiac magnetic resonance; CMR, cardiac magnetic resonance; EDV, end diastolic volume; EF, ejection fraction; EMB, endomyocardial biopsy; LBBB, left bundle branch block; LGE, late gadolinium enhancement; LV, left ventricle; RBBB, right bundle branch block; RV, right ventricle; RVOT, right ventricular outflow tract. Adapted from Corrado et al. [11].

	Criteria for RV Involvement	Criteria for LV Involvement
**I. Morpho-functional ventricular abnormalities**	***By 2D echocardiogram,******CMR or angiography:*** ***Major*** • Regional RV akinesia, dyskinesia or bulging plus one of the following: - global RV dilatation (increase of RV EDV according to the imaging test specific monograms for age, sex and BSA) *or* - global RV systolic dysfunction (reduction of RV EF according to the imaging test specific monograms for age and sex)	***By 2D echocardiogram,******CMR or angiography:*** ***Minor*** • Global LV systolic dysfunction (depression of LV EF or reduction of echocardiographic global longitudinal strain), with or without LV dilatation (increase in LV EDV according to the imaging test specific nomograms for age, sex, and BSA)
	***Minor*** • Regional RV akinesia, dyskinesia or aneurysm of RV free wall	***Minor*** • Regional LV hypokinesia or akinesia of LV free wall, septum or both
**II. Structural myocardial abnormalities**	***By CECMR:*** ***Major*** • Transmural LGE (stria pattern) of ≥1 RV region(s) (inlet, outlet, and apex in 2 orthogonal views)	***By CECMR:*** ***Major*** • LV LGE (stria pattern) of ≥1 Bull’s Eye segment(s) (in 2 orthogonal views) of the free wall (subepicardial or midmyocardial), septum or both (excluding septal junctional LGE)
	***By EMB (limited indications):*** ***Major*** • Fibrous replacement of the myocardium in ≥1 sample, with or without fatty tissue	
**III. ECG repolarization abnormalities**	***Major*** • Inverted T waves in right precordial leads (V_1_, V_2_ and V_3_) or beyond in individuals with complete pubertal development (in the absence of complete RBBB) ***Minor*** • Inverted T waves in leads V1 and V2 in individuals with completed pubertal development (in the absence of complete RBBB) • Inverted T waves in V1, V2, V3 and V4 in individuals with completed pubertal development in the presence of complete RBBB.	***Minor*** • Inverted T waves in left precordial leads (V_4_–V_6_) without complete LBBB
**IV. ECG depolarization abnormalities**	** *Minor* ** • Epsilon wave (reproducible low amplitude signals between end of QRS complex to onset of the T wave) in the right precordial leads (V1 to V3) • Terminal activation duration of QRS ≥ 55 ms measured from the nadir of the S wave to the end of the QRS, including R’, in V1, V2 or V3 (in the absence of complete RBBB)	***Minor*** • Low QRS voltages (<0.5 mV peak to peak) in limb leads (in the absence of obesity, emphysema or pericardial effusion)
**V. Ventricular arrhythmias**	***Major*** • Frequent ventricular extrasystoles (>500 per 24 h) or non-sustained or sustained ventricular tachycardia of LBBB morphology ***Minor*** • Frequent ventricular extrasystoles (>500 per 24 h) or non-sustained or sustained ventricular tachycardia of LBBB morphology with inferior axis (“RVOT pattern”)	***Minor*** • Frequent ventricular extrasystoles (>500 per 24 h) or non-sustained or sustained ventricular tachycardia with an RBBB morphology (excluding the “fascicular pattern”)
**VI. Family** **history/genetics**	** *Major* ** • ACM confirmed in a first-degree relative who meets diagnostic criteria • ACM confirmed pathologically at autopsy or surgery in a first-degree relative • Identification of a pathogenic or likely pathogenetic ACM mutation in the patient under evaluation ** *Minor* ** • History of ACM in a first-degree relative in whom it is not possible or practical to determine whether the family member meets diagnostic criteria • Premature sudden death (<35 years of age) due to suspected ACM in a first-degree relative • ACM confirmed pathologically or by diagnostic criteria in second-degree relative	

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
