# Peer review of "The 2020 “Padua Criteria” for Diagnosis and Phenotype Characterization of Arrhythmogenic Cardiomyopathy in Clinical Practice"

_jcm, 2022, doi:10.3390/jcm11010279_

Round 1
Reviewer 1 Report
The manuscript focuses on the proposed 2020 Padua criteria and provides 3 case examples of their application. There are some minor typos and English corrections required but overall it communicates the criteria logically and uses classical case examples which are important to recognise.
Please include the specific gene variants identified in the cases. Did the 23 year old (example 3) undergo predictive testing i.e. only for the DSP variant or testing of all ACM genes?
The LGE patterns described are important to recognise and a key message to readers. In my opinion the manuscript would benefit from a visual aid either cartoon to illustrate these patterns. The CMR images included in the case examples are of average quality.
Minor comments
Title – suggest change “the clinical practice” to “clinical practice”
Table 1 – please add a title to the legend e.g. Padua criteria for ACM.
CE-CMR – what does CE mean? Presume contrast enhanced. Please define when first used in text.
EMB – abbreviation needs defining
PVBs – abbreviation needs defining
Line 133 and 135 – “puberal” typo – correct to “pubertal”
Line 143 – interobservator – correct to “interobserver”
Line 201 – correct to “38-year old” and change “referred” to “reported”
Lines 203-206, 212 – remove “The” from the start of each sentence.
Line 271 – correct to “Her father’s”
Author Response
Please include the specific gene variants identified in the cases. Did the 23 year old (example 3) undergo predictive testing i.e. only for the DSP variant or testing of all ACM genes?
Reply: we thank the Reviewer for the comment. As suggested, we added the genetic information in the new revised version of the manuscript (page 9, line 267-268; page 10, line 284-285-286, 293-294-295)
The LGE patterns described are important to recognise and a key message to readers. In my opinion the manuscript would benefit from a visual aid either cartoon to illustrate these patterns. The CMR images included in the case examples are of average quality.
Reply: we thank the Reviewer for the comment, however, the visualization of cartoon with the different pattern of LGE may be too small for the figure structured to also visualize the clinical data. We improved the quality of the CMR images by enhancing the figure resolution
Minor comments
Title – suggest change “the clinical practice” to “clinical practice”
Reply: as suggested we modified the title in the new revised version.
Table 1 – please add a title to the legend e.g. Padua criteria for ACM.
Reply: as suggested we modified the legend in the new revised version (page 2, line 81).
CE-CMR – what does CE mean? Presume contrast enhanced. Please define when first used in text.
Reply: as suggested we specify the acronymous of CE-CMR used first in the Table 1 and in the corresponding legend (page 2, line 83)
EMB – abbreviation needs defining
Reply: as suggested we specify the acronymous of EMB used first on page 2, line 84.
PVBs – abbreviation needs defining
Reply: as suggested we specify the acronymous of PVBs used first on page 5, line 155.
Line 133 and 135 – “puberal” typo – correct to “pubertal”
Reply: as suggested we correct “puberal” to “pubertal” (page 5, line 134, 136).
Line 143 – interobservator – correct to “interobserver”
Reply: as suggested we correct “interobservator” to “interobserver” (page 5, line 144).
Line 201 – correct to “38-year old” and change “referred” to “reported”
Reply: as suggested we correct “38-year old” and “referred” instead “reported” (line 202).
Lines 203-206, 212 – remove “The” from the start of each sentence.
Reply: as suggested we correct removed “The” from the start of the sentence reported by the Reviewer (page 6, line 204-207; page 7, line 213).
Line 271 – correct to “Her father’s”
Reply: We thank the Reviewer for this suggestion, and we provided to modify the text (page 11, line 284).
Reviewer 2 Report
This is a very valuable work with high practical implication. I find very useful the flowchart for phenotypic characterization as well the 3 cases presented as examples.
I also appreciate the comparison between the 2010 ITF criteria and the 2020 Padua criteria, and the reasons for which they were changed.
I have some minor revisions:
- On page 5, rows 113, 125, 128 please replace EBM with EMB
- On figure 2 (please see the word doc):
- As it stated in the text, the CMR revealed a “wide peri-tricuspid aneurysm, with an extreme thinning of the wall”. I agree that the RV free wall is indeed very thin in the sub-tricuspidal area in this patient. But then, the bottom F figure shows a supposed LGE hyperenhancement of a rather thick RV wall. I suspect that what arrows indicate in the F bottom figure is in fact epicardial fat, as confirmed by the F upper figure. Moreover, the whiteish appearance of the RV free wall in figures E and bottom F may be explained by different Times of Inversion (TI) used for nulling the LV and RV myocardium, respectively. It is well known that RV and LV myocardium display different nulling inversion times. In other words, I don’t think that transmural LGE of the RV free wall can be diagnosed in this patient. However, authors did very well that they did not include this CE-CMR major structural criterion among the others in table G for this patient. I suggest deleting the RV transmural LGE part of the CMR interpretation from the text and the legend of the figure.
- There is fatty infiltration of the RV free wall in this patient. However, the arrows in the F upper figure are located very far from the actual localization of fatty infiltration, indicating the epicardial fat. I suggest moving the arrow where the yellow one is placed.
- In table G, I suggest stating the exact criteria related to the specific patient instead of enumerating again the general criteria.
- On page 10, row 261 please replace the “RVOT view” with “RV inflow-outflow view”
- In Figure 4:
I agree that the LGE pattern in this patient is very suggestive of ALVC. However it is noteworthy that in the short axis view, the subepicardial scar is visible only on the inferior and infero-lateral walls. No hyperenhancement is seen on the anterior and lateral walls in this short axis view. The figure may be misleading since the structural criteria says that the LV LGE must be identified “in 2 orthogonal views”. This is not the case in this patient. I suspect that what arrows indicate in the 2C and 4C longitudinal views (E and D) is normal pericardium. I would report only the inferior and infero-lateral LGE.

Author Response
I have some minor revisions:
- On page 5, rows 113, 125, 128 please replace EBM with EMB
Reply: as suggested we specify the acronymous of EMB and we replaced EBM with EMB (page 5, line 114, 126, 129).
- On figure 2 (please see the word doc):
As it stated in the text, the CMR revealed a “wide peri-tricuspid aneurysm, with an extreme thinning of the wall”. I agree that the RV free wall is indeed very thin in the sub-tricuspidal area in this patient. But then, the bottom F figure shows a supposed LGE hyperenhancement of a rather thick RV wall. I suspect that what arrows indicate in the F bottom figure is in fact epicardial fat, as confirmed by the F upper figure. Moreover, the whiteish appearance of the RV free wall in figures E and bottom F may be explained by different Times of Inversion (TI) used for nulling the LV and RV myocardium, respectively. It is well known that RV and LV myocardium display different nulling inversion times. In other words, I don’t think that transmural LGE of the RV free wall can be diagnosed in this patient. However, authors did very well that they did not include this CE-CMR major structural criterion among the others in table G for this patient. I suggest deleting the RV transmural LGE part of the CMR interpretation from the text and the legend of the figure.
Reply: we are very grateful to the Reviewer for the observation. In accordance with your suggestion, we modified the Figure 2 specifying that the most important finding was the presence of RV fatty infiltration in the area while we agree that the presence of RV LGE could not be established. We insert in the new version of the revised manuscript the new Figure 4 and the corresponding legend (page 8).
- There is fatty infiltration of the RV free wall in this patient. However, the arrows in the F upper figure are located very far from the actual localization of fatty infiltration, indicating the epicardial fat. I suggest moving the arrow where the yellow one is placed.
Reply: as previously described we modified the figure accordingly with your suggestion.
- In table G, I suggest stating the exact criteria related to the specific patient instead of enumerating again the general criteria.
Reply: In table G we reported only the criteria achieved by the specific patient, although for teaching purposes we elected to leave the exact definition of each criteria
- On page 10, row 261 please replace the “RVOT view” with “RV inflow-outflow view”
Reply: as suggested we replaced the “RVOT view” with “RV inflow-outflow view” (page 10, line 274).
- In Figure 4:
I agree that the LGE pattern in this patient is very suggestive of ALVC. However it is noteworthy that in the short axis view, the subepicardial scar is visible only on the inferior and infero-lateral walls. No hyperenhancement is seen on the anterior and lateral walls in this short axis view. The figure may be misleading since the structural criteria says that the LV LGE must be identified “in 2 orthogonal views”. This is not the case in this patient. I suspect that what arrows indicate in the 2C and 4C longitudinal views (E and D) is normal pericardium. I would report only the inferior and infero-lateral LGE
Reply: we really appreciate the Reviewer’s observation and we agree that two orthogonal views to confirm the presence of LGE. On this basis we modified the Figure 4 and the legend by removing the four-chamber view and adding instead the short axis view to appreciate the ring pattern. We agree with the Reviewer that the LV anterior wall is not involved and subsequently we deleted the arrow. We insert in the new version of the revised manuscript the new Figure 4 and the corresponding legend (page 12).
Reviewer 3 Report
The examples showcasing how to use the criteria are really helpful and clarify the criteria and algorithm for diagnosis .
Author Response
Thank-you for your favorable comments
This manuscript is a resubmission of an earlier submission. The following is a list of the peer review reports and author responses from that submission.